**Article** https://doi.org/10.1038/s41467-024-50356-3

# 1,3-Difunctionalization of [1.1.1]propellane through iron-hydride catalyzed hydropyridylation

Changha Kim [1,2], Yuhyun Kim[1,2] & Sungwoo Hong [1,2]

Current methodologies for the functionalization of [1.1.1]propellane primarily focus on achieving 1, 3-difunctionalized bicyclo[1.1.1]pentane or ring-opened cyclobutane moiety. Herein, we report an innovative approach for the 1, 3-difunctionalization of [1.1.1]propellane, enabling access to a diverse range of highly functionalized cyclobutanes via nucleophilic attack followed by ring opening and iron-hydride hydrogen atom transfer. To enable this method, we developed an efficient iron-catalyzed hydropyridylation of various alkenes for C − H alkylation of pyridines at the C4 position, eliminating the need for stoichiometric quantities of oxidants or reductants. Mechanistic investigations reveal that the resulting N-centered radical serves as an effective oxidizing agent, facilitating single-electron transfer oxidation of the reduced iron catalyst. This process efficiently sustains the catalytic cycle, offering significant advantages for substrates with oxidatively sensitive functionalities that are generally incompatible with alternative approaches. The strategy presented herein is not only mechanistically compelling but also demonstrates broad versatility, highlighting its potential for late-stage functionalization.

[1.1.1]Propellane, characterized by its unique ring-strained structure, has been extensively employed in synthesizing various molecules[1–5]. The synthesis of bicyclo[1.1.1]pentane (BCP) derivatives commonly entails the 1,3-difunctionalization of [1.1.1]propellane, utilizing its unique ring-strained structure via radical or ionic pathways[6–22]. Additionally, the formation of methylenecyclobutane moieties through the use of [1.1.1]propellane represents another important pathway, generally realized through ring-opening reactions enabled by transition metal carbenes[23–25] or cations[26–28]. Such cationic addition to propellane triggers a transformation that produces a 3-methylenecyclobutan-1-ylium cation. Despite these advancements, the exploration of 1,3-difunctionalized cyclobutane moieties remains an uncharted area (Fig. 1a, up). This shortfall arises primarily from two key challenges: i) the requirement to simultaneously functionalize both the cationic carbon and the alkene site after ring-opening, necessitating mild conditions to prevent side reactions, and ii) the difficulty in attaining

selective functionalization complicated by the similar reactivity profiles of [1.1.1]propellane and alkenes towards radical reactions[1–5,29,30].

While observations have shown protonation of propellane by acetic acid[26,28], advancements in acid-promoted chemical transformations remain limited. Our preliminary studies indicated that the intermediate, formed by the nucleophilic addition of alcohol to N-amidopyridinium salts[31,32], effectively fragments [1.1.1]propellane, leading to methylenecyclobutane through the 3-methylenecyclobutan-1-ylium cation (see Fig. 2). Inspired by these findings, we hypothesized that the 1,3-difunctionalization of propellane might be feasible through a two-fold mechanism: nucleophilic addition to the resulting cation[33–45], coupled with a simultaneous metal-hydride hydrogen atom transfer (MHAT) to the alkene moiety after ring opening[46–54] (Fig. 1a, down). This strategy underscores the potential of [1.1.1]propellane in facilitating rapid access to a variety of functionally complex cyclobutanes through an

[1]Department of Chemistry, Korea Advanced Institute of Science and Technology (KAIST), Daejeon, Korea. [2]Center for Catalytic Hydrocarbon Functionalizations, Institute for Basic Science (IBS), Daejeon, Korea. ✉e-mail: hongorg@kaist.ac.kr

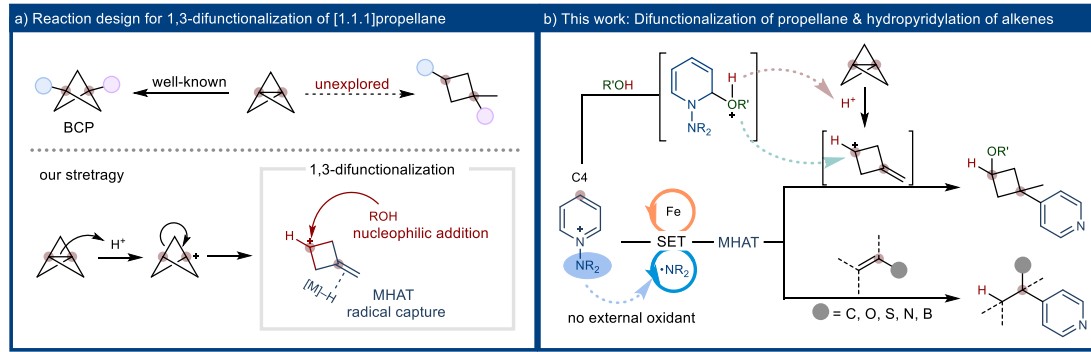

**Fig. 1 | Strategic design for [1.1.1]propellane 1,3-difunctionalization and regioselective olefin hydropyridylation. a** Reaction design for 1,3-difunctionalization of [1.1.1] propellane **b** Difuncitonalization of [1.1.1]propellane & hydropyridylation of alkenes.

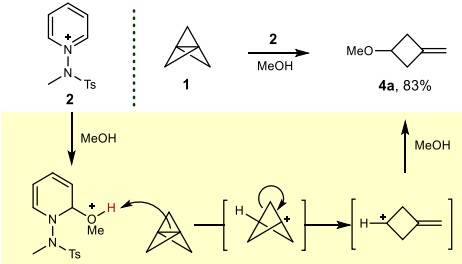

**Fig. 2 | Preliminary study.** [1.1.1]Propellane ring opening using N-amidopyridinium salt. Reaction conditions: **1** (0.075 mmol), **2a** (0.05 mmol) in MeOH (0.5 mL) at rt for 1 h under Ar atmosphere.

innovative three-component reaction that incorporates an array of alcohols and pyridine units. Based on this hypothesis, we refocused our efforts towards a hydropyridylation reaction, specifically tailored to react with the alkene moiety generated in-situ from [1.1.1] propellane.

There are two notable prior approaches to the radical hydropyridylation of alkenes via MHAT[55–61]. The Herzon group developed a cobalt-mediated radical hydropyridylation of alkenes using alkoxypyridinium salts and *tert*-butyl hydroperoxide (TBHP)[55,56]. However, employing both the stoichiometric Co and oxidant TBHP introduces complexity to the reaction, diminishing efficiency for large-scale applications. Additionally, achieving regioselective alkylation on unbiased pyridine cores is problematic due to competing sites (C2 vs C4) for radical interception. Recently, the Teskey group achieved photochemical hydropyridylation, involving dienes and pyridyl phosphonium salts with a cobalt-hydride catalyst[57]. However, due to the high reduction potential ($E_{red} = 1.51$ V vs SCE)[62] of phosphonium salts, this method requires a strong reductant such as the photoactivated Hantzsch ester ($E^*_{ox} = -2.28$ V vs SCE)[63], which restricts its compatibility with various functional groups. Our evaluation of the proposed three-component reactions revealed that these harsh conditions adversely affected the stability of the 3-methylenecyclobutan-1-ylium cation, underscoring the necessity for milder reaction conditions.

To address this challenge, we developed a strategy utilizing N-amidopyridinium salts as versatile reagents that facilitate both the [1.1.1]propellane ring-opening and hydropyridylation steps. We speculated that N-centered radicals derived from these salts[64–69] could serve as efficient oxidizing agents for the reduced state of the metal catalyst, thereby promoting the regeneration of the catalytic cycle without necessitating an external catalyst regenerator[65,69]. Subsequently, the alkyl radical intermediate, formed via iron-catalyzed HAT to the alkene, engages in a radical addition at the C4 position of N-amidopyridinium salts[70–73]. Our investigations revealed that these

mild conditions offer a suitable method for hydropyridylation of the in-situ generated 3-methylenecyclobutan-1-ylium cation, which is critical for the 1, 3-difunctionalization of [1.1.1]propellane. This approach facilitates a successful three-component reaction, providing swift access to a wide array of highly functionalized cyclobutanes and accommodating a diverse range of alkenes with various functional groups (Fig. 1b).

## Results
### Reaction discovery and optimization
Our preliminary results indicated that an acidic proton can be generated through the nucleophilic attack of alcohols on N-amidopyridinium salts. Focusing on the acid-mediated ring-opening of [1.1.1]propellane, we examined its reaction with N-amidopyridinium salt in MeOH. Notably, we observed a substantial conversion of propellane, which involved acid-catalyzed ring opening followed by nucleophilic addition of methoxide, leading to a new synthetic pathway to methylenecyclobutane **4a** (Fig. 2). Encouraged by these findings, our next goal is to combine olefin hydropyridylation with MHAT for the alkene moiety, laying the groundwork for an efficient 1, 3-difunctionalization strategy for [1.1.1]propellane.

To achieve this objective, we initially explored the development of an efficient MHAT method for alkenes, tailored to operate under mild conditions. This approach was specifically designed to be compatible with the unstable cation intermediate generated from [1.1.1]propellane. The characteristics of N-amidopyridinium salts and the efficiency of amidyl radicals are greatly influenced by the electronic and steric properties of the N-substituents. Therefore, we initially screened various N-substituents, with a focus on the in-situ generation of amidyl radicals and a catalyst regeneration strategy (Table 1).

Our screening identified N-amidopyridinium salts with a tosyl group as having the highest reactivity. Building on this, we further tested N-amidopyridinium salts **2a** and alkene **5a** (Supplementary Table 1). By employing Fe(acac)$_3$, we achieved an 82% yield in the hydropyridylated product **6a**, exhibiting exclusive regioselectivity for both the Markovnikov addition on the alkene and the C4 position of pyridine (entry 1). We observed that bulkier Fe(III) catalysts such as Fe(dpm)$_3$ and Fe(dibm)$_3$ were less efficient, likely due to hindered interaction with the amidyl radical (entry 2). Meanwhile, Fe(II) catalysts demonstrated moderate reactivity (entry 3), with a slight decomposition of N-amidopyridinium salt under the reaction conditions. Cobalt and manganese catalysts, although frequently used in MHAT reactions[46–48], showed less than 10% reactivity, making them unsuitable for this reaction (entry 4). When testing different alcohols, MeOH exhibited reactivity comparable to EtOH, while $^i$PrOH showed a decrease in reactivity, attributed to lower salt solubility (entry 5). The employment of PhSiH$_3$ led to a modest reduction in reactivity, achieving a yield of 58% (entry 6)[74]. The presence of oxygen, which

## Table 1 | Optimization of the Reaction Conditions[a]

| Entry | Reaction partner | Deviation from standard conditions | Yield 3a (%)[b] | Yield 6a (%)[b] |
|---|---|---|---|---|
| 1 | 5a | none | - | 77 (82) |
| 2 | 5a | Fe(dpm)$_3$ / Fe(dibm)$_3$ instead of Fe(acac)$_3$ | - | 38 / 56 |
| 3 | 5a | Fe(acac)$_2$ / Fe(OTf)$_2$ instead of Fe(acac)$_3$ | - | 55 / 6 |
| 4 | 5a | Co(acac)$_2$ / Co(salen) / Mn(acac)$_3$ instead of Fe(acac)$_3$ | - | 9 / trace / trace |
| 5 | 5a | MeOH / $^i$PrOH instead of EtOH | - | 70 / 35 |
| 6 | 5a | PhSiH$_3$ instead of Ph(O$^i$Pr)SiH$_2$ | - | 58 |
| 7 | 5a | O$_2$ / Air condition instead of Ar condition | - | trace / 39 |
| 8 | 5a | without Fe catalyst / silane | - | n.d. / n.d. |
| 9 | 5a | without base | - | 43 |
| 10 | 1 | none | 73 (d.r. = 3.0:1) | - |
| 11 | 1 | Fe(dpm)$_3$ instead of Fe(acac)$_3$ | 57 (d.r. = 2.5:1) | - |
| 12 | 1 | NaOAc instead of NaHCO$_3$ | 8[c] | - |
| 13 | 1 | without base | 46 (d.r. =2.8:1) | - |

[a]Reaction conditions: **2a** (0.1 mmol), **5a** or **1** (0.15 mmol), NaHCO$_3$ (0.1 mmol), Ph(O$^i$Pr)SiH$_2$ (0.2 mmol), Fe(acac)$_3$ (5 mol%) in EtOH (1.0 mL) at rt for 15 h under Ar. [b]Yields were determined by $^1$H NMR spectroscopy, and 1,1,2,2-tetrabromoethane was used as an internal standard. [c]32% of hydropyridylated BCP product was obtained. Isolated yield in parenthesis. n.d. = not detected.

typically aids in Fe catalyst regeneration[75,76], severely impeded the reaction at high concentrations, leading to only trace amounts of the desired product. Likewise, a substantial decrease in reactivity was also observed under air conditions (entry 7). Control experiments conducted without the Fe catalyst or silane failed to produce any hydropyridylated product, underscoring the essential roles of these components (entry 8). The fact that the reaction proceeds in the absence of an external base, achieving a 43% yield (entry 9), suggests that the in-situ generated amide or pyridine may serve as the base. Building upon hydropyridylation reaction conditions, we then continue the optimization of the 1, 3-difunctionalization of [1.1.1]propellane, focusing on reactions involving the 3-methylenecyclobutan-1-ylium cation. The reaction with EtOH as solvent and reactant, 1, 3-difunctionalized cyclobutane product **3a** was obtained with 73% yield (d.r. = 3.0: 1) (entry 10)[77]. Even when using the Fe catalyst with a bulky ligand, there is little change in diastereoselectivity, suggesting that the Fe catalyst has no influence on the radical insertion process of the N-amidopyridinium salt (entry 11). During the optimization process, we noted variations in MHAT selectivity (alkene vs. [1.1.1]propellane) contingent upon the base used. For example, employing NaOAc as the base resulted in predominantly the formation of hydropyridylated BCP through direct MHAT of [1.1.1]propellane (entry 12). Similar to the hydropyridylation, a slight decrease in reactivity was observed in the absence of base (entry 13).

### Substrate scope studies

Leveraging these optimized conditions, we initiated a comprehensive investigation into the scope of alcohols, particularly emphasizing those commonly utilized as solvents (Fig. 3). Our findings indicate that primary alcohols exhibit outstanding reactivity under these conditions (**3a, b**). Expanding our scope to include secondary alcohols, we encountered solubility issues with the N-amidopyridinium salt, necessitating the introduction of a co-solvent system with dichloromethane (DCM). To our delight, in this slightly modified system, both acyclic (**3c**) and cyclic (**3d**) secondary alcohols displayed robust reactivity. Although tertiary alcohols exhibited a decreased reactivity, a consequence of their reduced nucleophilicity, they nonetheless

participated in the reaction with reasonable efficacy (**3e, f**). Notably, the method showed remarkable compatibility with structurally intricate alcohol solvents, such as ethylene glycol ether, maintaining efficient reaction progress (**3g, h**). Moreover, the versatility of the pyridine core was explored using pyridines with various substituents at the C2 or C3 positions. A diverse array of pyridines, including aryl (**3i–3k**), heteroaryl (**3l**), trifluoromethyl (**3m**), cyano (**3n**), methyl (**3o–q**)substituted variants, consistently yielded 1, 3-difunctionalized products. In addition, the reaction also efficiently accommodated unsubstituted pyridine (**3r**), achieving effective ring opening and demonstrating the method's broad tolerability. Pleasingly, 1,3-difunctionalization was also applicable to pyridine-based pharmaceuticals such as vismodegib (**3s**), bisacodyl (**3t**), and pyriproxyfen (**3u**) moieties.

Building upon our understanding of the 1, 3-difunctionalization reaction, we speculated that the use of a catalytic amount of N-amidopyridinium salt would facilitate the synthesis of methylenecyclobutanes featuring a broad range of alcohol integrations. The study yielded impressive results: primary (**4a, b**), secondary (**4c**), and tertiary alcohols (**4d**), as well as ethylene glycol ether (**4e**), all showed significant reactivity. Notably, the reaction involving benzyl alcohol stood out, delivering a high yield and highlighting its potential as a preferred reactant for methylenecyclobutane production (**4f**). This methodology has proven to be highly effective, producing an array of ether-containing methylenecyclobutanes. Collectively, these findings highlight the robustness and versatility of method, paving the way for diverse applications in the realm of complex molecule synthesis.

We confirmed during the optimization process that this method is efficiently applicable to olefins as well (Table 1, entry 1). Particularly, unlike previous hydropyridylation[55–58], which required an external catalyst regenerator, we believed it could be applied to various alkenes, ranging from unactivated to activated ones. Therefore, we endeavored to establish the generality of the hydropyridylation using **2a** as the model substrate (Fig. 4). Initially, our focus was on evaluating the reactivity of 1, 1-disubstituted alkenes. We observed successful outcomes with both acyclic (**6a, b**, and **e**) and cyclic (**6c** and **6d**) alkenes. Notably, alkenes containing amide (**6f**), ester (**6g**), and ketone

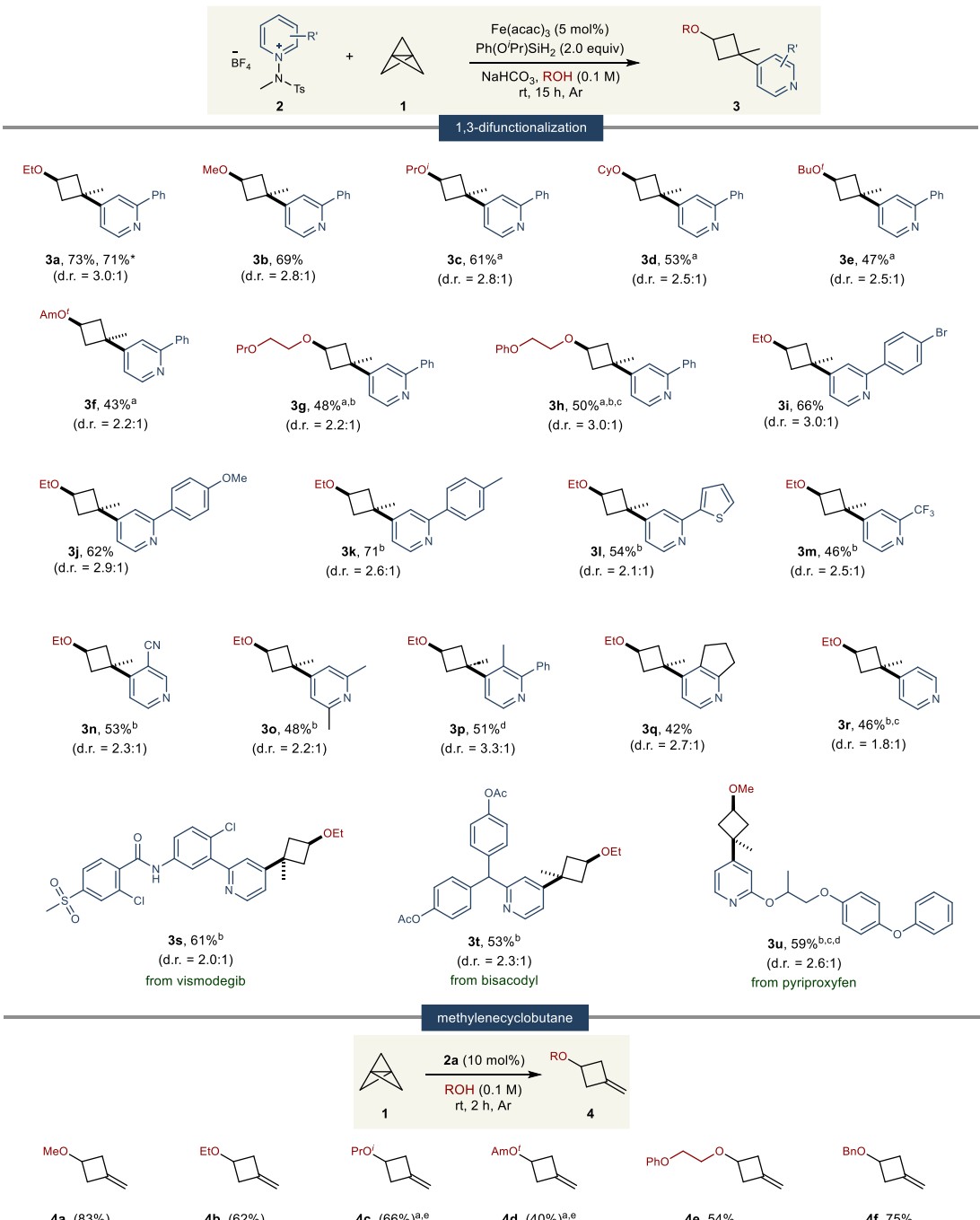

**Fig. 3 | Scope of 1,3-difunctionalization of [1.1.1]propellane and methylenecyclobutane.** Reaction conditions of 1,3-difunctionalization: **2** (0.1 mmol), **1** (0.15 mmol), NaHCO₃ (0.1 mmol), Ph(OⁱPr)SiH₂ (0.2 mmol), Fe(acac)₃ (5 mol%) in alcohol (1.0 mL) at rt under Ar atmosphere. Reaction conditions of methylenecyclobutane: **1** (0.1 mmol), **2a** (0.01 mmol) in alcohol (1.0 mL) at rt for 2 h under Ar atmosphere. [a]Using cosolvent system (DCM:ROH = 1:1). [b]Using 10 mol% of Fe(acac)₃. [c]3.0 equivalent of **1** was used. [d]Using NaF instead of NaHCO₃. [e]Using 1.0 equiv of **2a**. [*]1.0 mmol scale reaction. Isolated yields. NMR yields of volatile products in parenthesis. Diastereomeric ratios were determined from the ¹H NMR of crude mixtures.

(**6h**) groups also exhibited commendable reactivity, offering opportunities for further functionalization[78–80]. This method proved applicable to mono-substituted alkenes, encompassing a simple carbon chain (**6i**), aryl group (**6j** and **6k**), and hydroxy group (**6l**). The reaction conditions were sufficiently mild to preserve sensitive functional groups such as carboxylic acid (**6m**), Boc-protected amine (**6n**), urea (**6o**), cyano (**6p**), trimethylsilane (**6q**), and alkyne (**6r**) intact. For the ester group, both aryl and alkyl esters (**6s** and **6t**) were compatible. Additionally, alkyl halides, including bromine (**6u**) and chlorine (**6v**), proceeded smoothly in hydropyridylation. In diene cases,

hydropyridylation preferentially occurred at one of the alkene sites, allowing further functionalization of the remaining alkene (**6w**). When presented with 1,1-disubstituted and terminal alkenes, the hydropyridylation preferentially targeted the 1,1-disubstituted alkene, reflecting the differential MHAT rates of the Fe catalyst (**6x**). Further, we assessed the reactivity of internal alkenes[61,81–84], observing good outcomes with cyclic alkenes regardless of ring size (**6y–ab**). We successfully accomplished hydropyridylation across a spectrum of alkenes, encompassing 1,2-disubstituted, tri-substituted, and tetra-substituted varieties (**6ac–ah**). Notably, even an aldehyde, which

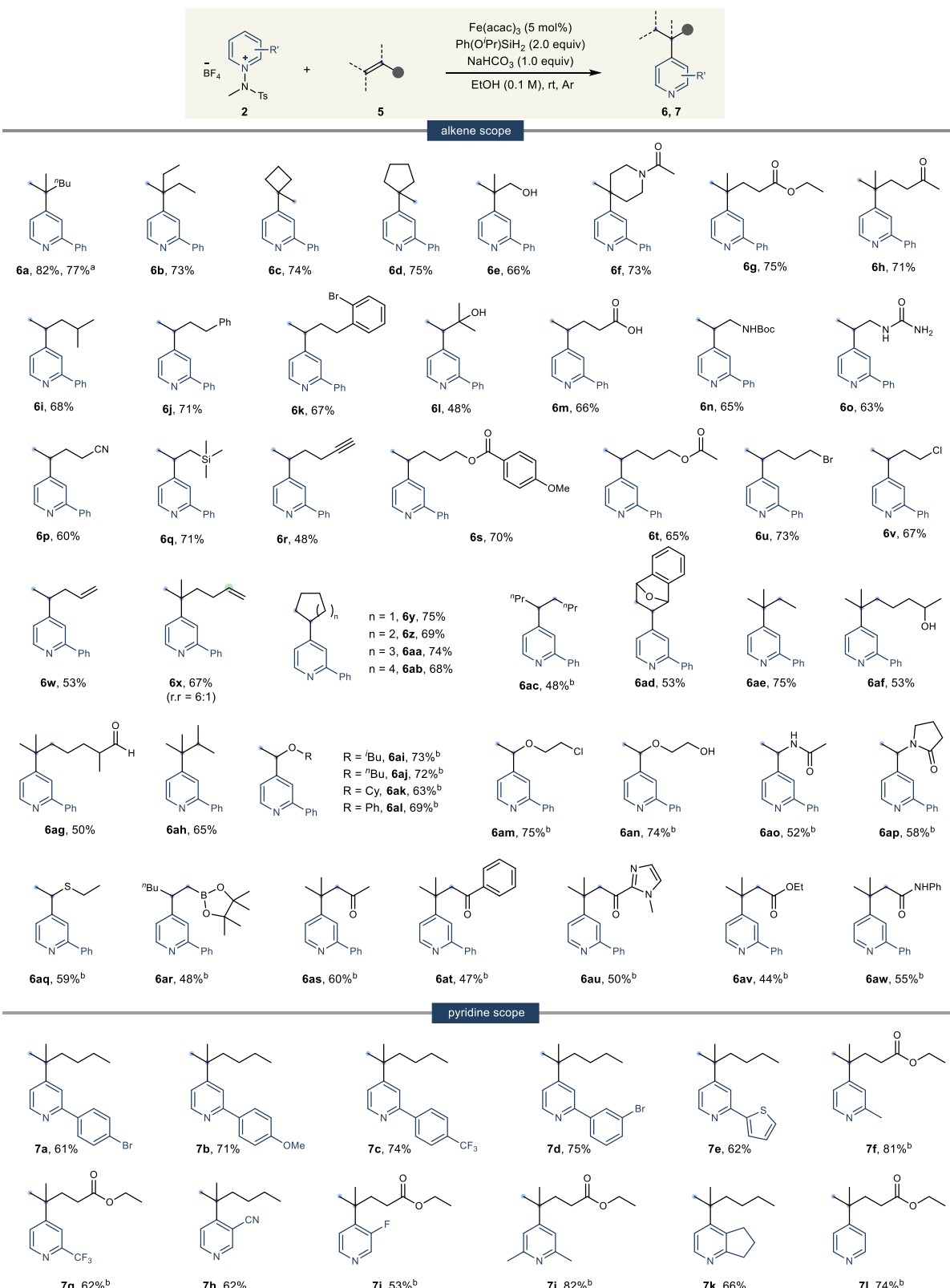

**Fig. 4 | Scope of C4/Markovnikov regioselective hydropyridylation of alkene.**
Reaction conditions: **2** (0.1 mmol), **5** (0.15 mmol), NaHCO₃ (0.1 mmol), Ph(OⁱPr)SiH₂ (0.2 mmol), Fe(acac)₃ (5 mol%) in EtOH (1.0 mL) at rt under Ar atmosphere.

[a]1.0 mmol scale reaction. [b]Using 10 mol% of Fe(acac)₃. Isolated yields. Regioselectivity was determined by ¹H NMR.

contains a proton which can be easy to undergo HAT, exhibited modest reactivity (**6ag**)[85–87]. Our exploration extended to activated alkenes as well, where heteroatom-substituted alkenes, including vinyl ether (**6ai–an**), enamine (**6ao** and **ap**), vinyl sulfide (**6aq**), and alkenyl

boronic ester (**6ar**), yielded the desired products with excellent regioselectivity. Significantly, while MHAT utilizing an Fe catalyst is typically associated with electron-rich alkenes, our hydropyridylation approach demonstrated notable efficacy in the functionalization of

**Fig. 5 | Late-stage functionalization of complex molecules.** [a]Using 10 mol% of Fe(acac)₃. [b]Using MeOH solvent instead of EtOH. Isolated yields. Diastereomeric ratios were determined by ¹H NMR.

α,β-unsaturated alkenes to afford β-functionalized carbonyl compounds (**6as–aw**). This offers an alternative mild method for Michael-type reactions involving organometallic reagents. Closely examining the pyridine scope, we found that our methodology operates smoothly in hydropyridylation reactions. Aryl (**7a–d**), heteroaryl (**7e**), methyl (**7f**), trifluoromethyl (**7g**), cyano (**7h**), and fluoro (**7i**) groups all exhibited excellent tolerance. Impressively, not only mono-substituted pyridines but also 2, 6 or 2, 3-disubstituted pyridines demonstrated notable reactivity (**7j–k**). Crucially, the use of unsubstituted N-amidopyridinium salt resulted in the synthesis of the desired product with exclusive C4 and Markovnikov regioselectivity (**7l**).

To further demonstrate the practical applicability of our developed methodology, we conducted a series of experiments focused on the late-stage functionalization of complex bioactive molecules (Fig. 5). We were pleased to discover that our method efficiently transformed alkene derivatives derived from fenofibrate (**8a**), zaltoprofen (**8b**), tyrosine (**8c**), and D-glucal (**8e**) into their corresponding hydropyridylated products under standard conditions. Notably, even eugenol (**8d**), which possesses a redox-active phenol group, yielded the desired product satisfactorily. The versatility of our approach was further highlighted by its successful application in the functionalization of structurally complex molecules, including steroids and terpenoids, exemplified by estrone (**8f**), boldenone undecylenate (**8g**), and citronellol (**8h**). Additionally, our method exhibited remarkable selectivity in modifying complex pyridine moieties (**8i–k**). These results significantly expand the scope of our synthetic method, underscoring its potential for the selective modification of pharmacologically relevant compounds.

### Control experiments and proposed mechanism

To gain detailed insights into the mechanism of 1, 3-difunctionalization of [1.1.1]propellane and hydropyridylation, various mechanistic studies were conducted (Fig. 6). Initially, to ascertain the role of the N-amidopyridinium moiety in facilitating ring opening, we performed control experiments. These experiments demonstrated that, in the presence of N-amidopyridinium salt **2a**, the methylenecyclobutane product **4e** was synthesized with a 54% yield. Conversely, the absence of N-amidopyridinium salt or the inclusion of NaBF₄ as an additive resulted in only trace yields of product 4e. This finding prominently highlights the critical role of the N-amidopyridinium salt in enabling the ring opening of [1.1.1]propellane, as illustrated in Fig. 6a. Then, deuterium labeling experiments were carried out to elucidate the ring-opening step of [1.1.1]propellane. Upon changing the reaction solvent MeOH to CD₃OD, 99% deuterium incorporation was observed at the oxygen α-position of the product (Fig. 6b). This result confirmed that the acidic proton used in the ring-opening step is derived from the proton in the alcohol solvent. When the reaction was conducted with a C4 blocked pyridinium salt **2n**, we observed the formation of methylenecyclobutane **4b**; however, the product resulting from insertion at C2, **3v**, was not detected (Fig. 6c). This indicates that the N-substituent effectively inhibits radical insertion at the C2 position. Next, in the hydropyridylation of the alkene, we sought to confirm the involvement of a radical pathway through MHAT. The reaction was conducted under optimized conditions using vinylcyclopropane (**5ax**) and hepta-1,6-diene (**5ay**) (Fig. 6d). The results revealed the occurrence of both ring opening (**6ax**) and ring closing (**6ay**), indicating the generation of a Markovnikov-selective radical. Furthermore, when the radical scavenger TEMPO was added as an additive and the reaction proceeded, the desired product was not formed at all, and alkyl radical-TEMPO adducts were observed in the HRMS (Fig. 6e).

On the basis of these mechanistic studies, we propose a mechanism for the 1,3-difunctionalization of [1.1.1]propellane and hydropyridylation of alkenes (Fig. 6f). The nucleophilic attack of alcohol on N-amidopyridinium salt **A** generates the acidic oxonium cation **B** (Part A)[88]. The propellane is subsequently protonated by the acidic proton of oxonium cation **B**, triggering a rearrangement that yields unstable cation **D**. Concurrently, the generated cation **D** is subjected to alcohol insertion, culminating in the formation of intermediate **E**[33–45]. Following this, intermediate **E** may partake in a proton exchange with intermediate **C** or initiate a chain reaction pathway by protonating [1.1.1]propellane, ultimately yielding methylenecyclobutane **F**. Control experiments with propellane revealed that methylenecyclobutane **F** is produced in ~9% yield with Fe(II) or Fe(III) (See the SI for details), even when N-amidylpyridinium salt is not present. This

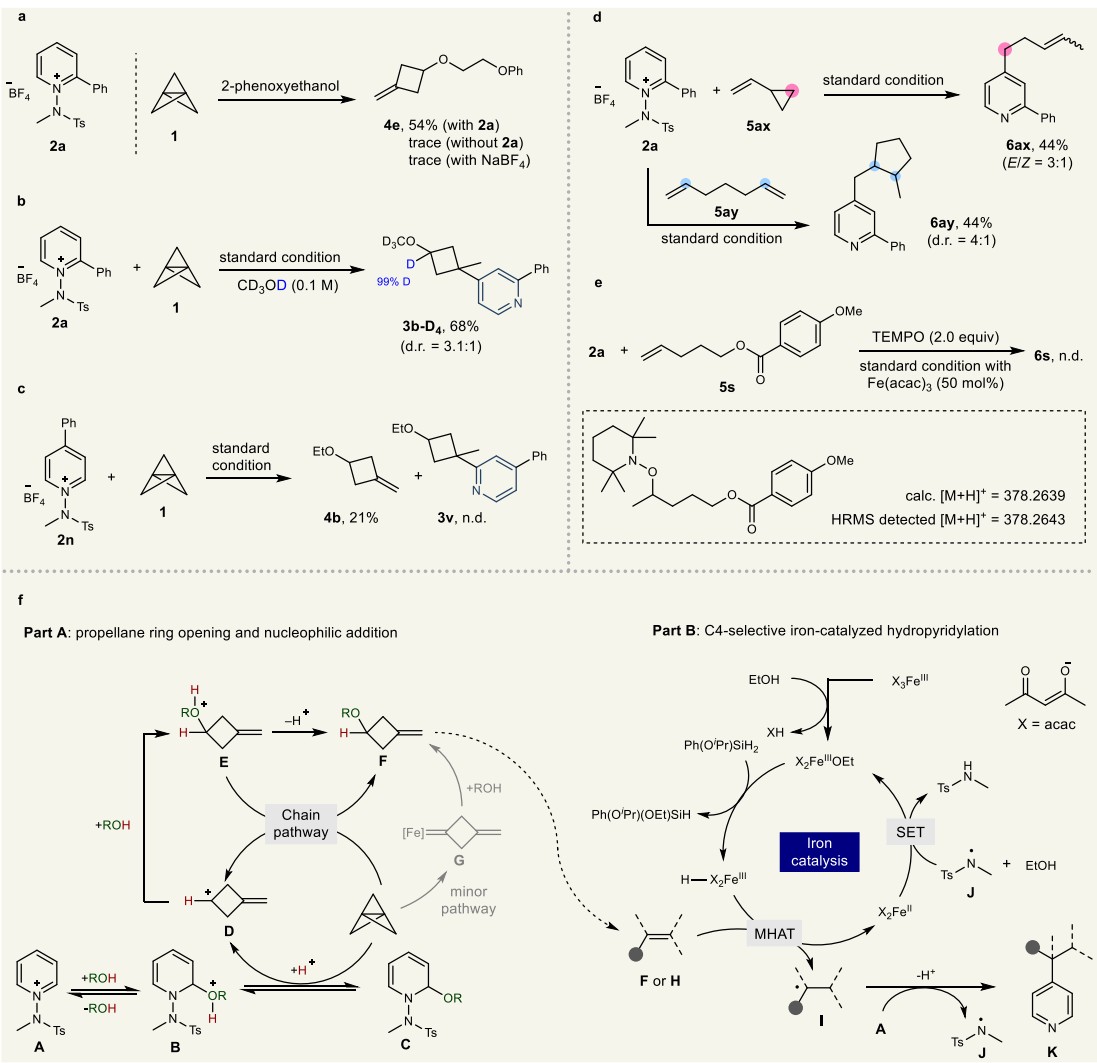

**Fig. 6 | Experimental mechanistic investigations and proposed mechanism. a** Control experiment of [1.1.1]propellane ring opening. **b** Deuterium labeling experiment. **c** C4-blocked pyridine experiment. **d** Radical clock experiment. **e** Radical capture experiment. **f** Proposed mechanism.

result indicates that the pathway culminating in the iron carbene complex **G** likely functions as a minor route in these reactions. The methylenecyclobutane **F** is subsequently utilized in the ongoing hydropyridylation process (Part B). During this phase, the FeX$_3$(III) undergoes solvolysis with alcohol, leading to the formation of the activated FeX$_2$OEt(III) catalyst[74]. Following this, the hydride/alkoxy exchange with silane gives rise to the formation of FeX$_2$H(III). The iron hydride then interacts with alkenes **F** or **H** via MHAT, generating the Markovnikov-selective alkyl radical intermediate **I**. This alkyl radical **I** undergoes radical insertion into the C4 position of N-amidopyridinium salt **A**, followed by deprotonation, to yield the desired product **K** along with the formation of the amidyl radical **J**. The amidyl radical **J** then acts as an oxidant, facilitating the regeneration of FeX$_2$OEt(III) from FeX$_2$(II), thereby completing the catalytic cycle.

## Discussion

In summary, we have successfully developed a highly efficient, iron-hydride catalyzed Markovnikov/C4 selective hydropyridylation of alkenes, operable under mild conditions. Our groundbreaking technique overcomes previous challenges by leveraging N-centered radicals derived from N-amidopyridinium salts as efficient oxidizing agents. This innovation not only streamlines the catalytic cycle but also exhibits broad compatibility with various functional groups, including

those sensitive to oxidation. Significantly, we have adeptly extended this hydropyridylation method to the 1, 3-difunctionalization of [1.1.1] propellane, facilitating the synthesis of intricate cyclobutanes via a three-component reaction. This method integrates diverse alcohols and pyridines, showcasing a distinctive mechanistic pathway and underscoring its potential for late-stage functionalization of complex, biologically relevant molecules. Overall, this study presents a robust, versatile, and sustainable methodology, setting the stage for transformative advancements in the functionalization of alkenes and [1.1.1] propellane.

## Methods

### Representative procedure for the 1, 3-difunctionalization of [1.1.1]propellane

To a 50 mL round-bottom flask equipped with a magnetic bar, N-amidopyridinium salt **2a** (1.0 equiv, 1.0 mmol, 426.1 mg), sodium bicarbonate (1.0 equiv, 1.0 mmol, 84.0 mg), and iron catalyst Fe(acac)$_3$ (5 mol%, 0.05 mmol, 17.7 mg) were added. The flask was then sealed with a rubber septum, evacuated, and back-filled with argon. EtOH solvent (0.1 M, 10.0 mL) and Ph(O$^i$Pr)SiH$_2$ (2.0 equiv, 2.0 mmol, 0.36 mL) were added, followed by [1.1.1]propellane **1** (1.0 M in benzene, 1.5 equiv, 1.5 mmol, 1.5 mL) via syringe. The reaction mixture was stirred at room temperature for 24 h. After reaction completion, the

reaction mixture was diluted with ethyl acetate, washed with water, and extracted ethyl acetate three times. After removal of solvent, the residue was purified by flash column chromatography on silica gel (eluent: ethyl acetate/*n*-hexane = 1:10) to give the desired product **3a** (71%, 189.1 mg).

## Representative procedure for the hydropyridylation of alkene

To a 50 mL round-bottom flask equipped with a magnetic bar, N-amidopyridinium salt **2a** (1.0 equiv, 1.0 mmol, 426.1 mg), sodium bicarbonate (1.0 equiv, 1.0 mmol, 84.0 mg), and iron catalyst Fe(acac)$_3$ (5 mol%, 0.05 mmol, 17.7 mg) were added. The flask was then sealed with a rubber septum, evacuated, and back-filled with argon. EtOH solvent (0.1 M, 10.0 mL) and Ph(O$^i$Pr)SiH$_2$ (2.0 equiv, 2.0 mmol, 0.36 mL) were added, followed by 2-methyl-1-hexene **5a** (1.5 equiv, 1.5 mmol, 0.21 mL) via syringe. The reaction mixture was stirred at room temperature for 15 h. After reaction completion, the reaction mixture was diluted with ethyl acetate, washed with water, and extracted ethyl acetate three times. After the removal of the solvent, the residue was purified by flash column chromatography on silica gel (eluent: ethyl acetate/n-hexane = 1:10) to give the desired product **6a** (77%, 194.5 mg).

## Data availability

Experimental procedure and characterization data of new compounds are available within Supplementary Information. All data are available from the corresponding author. This material is available free of charge via the Internet.

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

## Acknowledgements

This research was supported financially by the Institute for Basic Science (IBS-R010-A2).

## Author contributions

C.K. and Y.K. performed the experiments. S.H. directed and supervised the project. All authors contributed to the preparation of the manuscript.

## Competing interests

The authors declare no competing interests.
