## [Peer Review File · Nature Communications]

1,3-Difunctionalization of [1.1.1]Propellane through Iron-Hydride Catalyzed HydropyridylationReviewers' Comments:

Reviewer #1:

Remarks to the Author:

Compared with the conventional 1,3-difunctionalization of [1.1.1]propellane, the exploration of 1,3-difunctionalized cyclobutane moieties derived from [1.1.1]propellanes are largely unexplored. In this manuscript, Hong and co-workers describe a generally applicable and iron-hydride catalyzed hydroxyridylation of various alkenes under practical and external oxidant-free conditions. The title reactions tolerate a broad range of alkenes ranging from unactivated to activated ones, allowing the straightforward access to diverse hydroxyridylation compounds. N-Amidopyridinium salts served both as pyridine scaffolds and N-radical sources, and the latter has proven to be oxidizing agents to complete the catalytic cycle. Successful application of this methodology to late-stage modification of bioactive compounds also highlights its synthetic potential. Mechanistic studies including supported the proposed mechanism.

Moreover, the manuscript and SI file are of high quality. In conclusion, I strongly recommend the acceptance of this work after addressing several minor points.

1) Typos:

Page 3, Fig. 1a, 'MHAT radical caputre' should be 'MHAT radical caputre'

Page 4, line 84, 'N-substitutens' should be 'N- substituents'

2) Page 8, line 165, '...which can be easy to HAT' might add a verb after 'to', such as 'undergo'.

3) Page 13, in the sentence 'Representative procedure for the migratory hydroamination of unactivated alkenes', the description of 'hydroamination' is not correct, which should be 'hydroxyridylation'.

4) The format should be carefully checked throughout the whole manuscript.

Page 1, line 13, the word 'N' in the sentence 'N-centered radical...' should be italic. And, similar mistake, please see: Page 13, Line 251.

Reviewer #2:

Remarks to the Author:

The manuscript by Hong et al. described the iron-catalyzed hydroxyridylation of alkenes and [1.1.1]propellane, synthesizing 4-substituted pyridines. The reaction requires 2 equiv of phenylisopropyl silane, which is a compound of limited commercial availability, the same is true for the propellane, whose preparation is not even suitable for a student's lab. They described basically two different substrates in the manuscript and from that aspect the title of the paper doesn't make much sense, because it attempts to convey an inner connection between these substrates. In fact, the authors described one standard catalytic system for two different substrates. Reaction conditions are in general mild, however, the N-amidopyridinium salts have to be made and the authors do not disclose if there is a chance of recycling of the remains after the reaction. The yields are in general in the range of 43-73% for the propellane hydroxyridylation and 44-82% for the olefin hydroxyridylation. The authors also follow the recent trend and include examples for late-stage functionalization using their catalyst system, resulting in good yields in most cases. Certainly the mild reaction conditions are helpful here. The authors also investigated the nature of the reaction towards the inclusion of radicals and presented proof for this assumption (trapping, radical clock). How do the authors explain that the "iron carbene" pathway is the minor pathway? And is there any methodology to promote this pathway, which is proposed to only give a small conversion to the exocyclic cyclobutene product?

Did the authors have data from calculations that would establish this carbene route? Another question deals with the Table S2: in the optimization of the Markovnikov addition the authors investigated different amounts of iron catalyst (5-20 mol%) without any effect on the yield. However, increase in alkene does raise the yield. What is the explanation for that?

The representative procedure given in the manuscript should describe a more practical size scale (at least 0.5-1 mmol) and include all the amounts in mg of the reagents and substrates instead of only give the product amount.

The methodology appears to be broadly applicable, leading to a series of functionalized pyridines, including cyclobutane-derived ones. The manuscript therefore is interesting to the organic synthetic community and publication as a research paper can be recommended.

Reviewer #3:

Remarks to the Author:

Hong described iron-hydride catalyzed C–H alkylation of pyridines at the C4 position for the difunctionalization of [1.1.1]propellane and hydroxyarylation of alkenes. This protocol is synthetic useful. The optimization of reaction conditions is systematically demonstrated. The substrate scope is well studied. The catalytic pathway is well designed. The experimental procedures and characterization data are clean and well organized. In my opinion, this manuscript could be published in Nature Commun after the following minor revisions have been addressed.

The title, introduction section and Fig. 1 confused me. Although the sections show the reactions “the difunctionalization of [1.1.1]propellane and hydroxyarylation of alkenes”, it is hard to grasp the essence. Please rewrite the sections. Which reaction is more important in this manuscript?

Fig 4, 6ar-6aw; Is the regioisomer observed? If not, the author should discuss the reason.

Table 1, Can the author decrease the amount of alcohol? How is about other nucleophiles such as phenol or amines?

Fig 3, The use of bulky alcohol is nice. However, the yield of bulky alcohol is not excellent. Can the author improve it?

Reviewer #1 (Remarks to the Author):

Compared with the conventional 1,3-difunctionalization of [1.1.1]propellane, the exploration of 1,3-difunctionalized cyclobutane moieties derived from [1.1.1]propellanes are largely unexplored. In this manuscript, Hong and co-workers describe a generally applicable and iron-hydride catalyzed hydroxyridylation of various alkenes under practical and external oxidant-free conditions. The title reactions tolerate a broad range of alkenes ranging from unactivated to activated ones, allowing the straightforward access to diverse hydroxyridylation compounds. N-Amidopyridinium salts served both as pyridine scaffolds and N-radical sources, and the latter has proven to be oxidizing agents to complete the catalytic cycle. Successful application of this methodology to late-stage modification of bioactive compounds also highlights its synthetic potential. Mechanistic studies including supported the proposed mechanism.

Moreover, the manuscript and SI file are of high quality. In conclusion, I strongly recommend the acceptance of this work after addressing several minor points.

1) Typos:

Page 3, Fig. 1a, 'MHAT radcial caputre' should be 'MHAT radical caputre'

Page 4, line 84, 'N-substitutens' should be 'N- substituents'

2) Page 8, line 165, '...which can be easy to HAT' might add a verb after 'to', such as 'undergo'.

3) Page 13, in the sentence 'Representative procedure for the migratory hydroamination of unactivated alkenes', the description of 'hydroamination' is not correct, which should be 'hydroxyridylation'.

4) The format should be carefully checked throughout the whole manuscript.

Page 1, line 13, the word 'N' in the sentence 'N-centered radical...' should be italic. And, similar mistake, please see: Page 13, Line 251.

Response and Action: We thank the reviewer for their careful evaluation of our work and their supportive feedback. All the indicated errors have been corrected in the revised manuscript as suggested by the reviewer.

Reviewer #2 (Remarks to the Author):

The manuscript by Hong at al. described the iron-catalyzed hydroxyridylation of alkenes and [1.1.1]propellane, synthesizing 4-substituted pyridines. The reactiuon requires 2 equiv of phenylisopropyl silane, which is a compound of limited commercial availability, the same is true for the propellane, whose preparation is not even suitable for a students lab. They described basically two different substrates in the manuscript and from that aspect the titel of the paper doesn't make much sense, because it attempts to convey an inner connection between these substrates. In fact, the authors described one standard condition catalytic system for two different substrates. Reaction conditions are in general mild, however, the N-amidopyridinium salts have to be made and the authors do not disclose if there is a chance of recycling of the remains after the reaction. The yields are in general in the range of 43-73% for the propellane hydroxyridylation and 44-82% for the olefin hydroxyridylation. The authors also follow the recent hip trend and include examples for late-stage functionalization using their catalyst system, resulting in good yields in most cases. Certainly the mild reaction conditions are helpful here. The authors also investigated the nature of the reaction towards the inclusion of radicals and presented proof for this assumption (trapping, radical clock).

The methodology appears to be broadly applicable, leading to a series of functionalized pyridines, including cyclobutane-derived ones. The manuscript therefore is interesting to the organic synthetic community and publication as a research paper can be recommended.

Response and Action: We thank the reviewer for their careful evaluation of our work and their supportive feedback. Following the suggestion, we have revised the title to "1,3-Difunctionalization of [1.1.1]Propellane through Iron-Hydride Catalyzed Hydroxyridylation" to better reflect the focus of our manuscript. Accordingly, we have updated the content of abstract and the introduction section to clarify

the essence of the reactions discussed. We emphasize that the distinctive 1,3-difunctionalization of [1.1.1]propellane for the synthesis of an array of 1,3-difunctionalized cyclobutanes is the primary focus of our study, with the Fe-catalyzed Markovnikov hydroxyarylation of various alkenes under mild reaction conditions serving as another significant aspect. We hope these revisions address your concerns and improve the clarity of our manuscript. Regarding [1.1.1]propellane, its ready availability has led to widespread use and ongoing research in numerous laboratories, underscoring the need to further explore the reactivity of this substrate. For *N*-amidopyridinium salts, despite requiring a three-step synthesis, these salts can be easily prepared on a large scale without column chromatography. In this study, *N*-amidopyridinium salts serve as the limiting reagent. For valuable materials, any remaining quantity is recycled.

- How do the authors explain that the "iron carbene" pathway is the minor pathway? And is there any methodology to promote this pathway, which is proposed to only give a small conversion to the exocyclic cyclobutene product?

- Did the authors have data from calculations that would establish this carbene route?

Response and Action: We thank the reviewer for pointing this out. Since the two questions are related, I will provide a combined answer for both. In references 23 and 25, ring opening of [1.1.1]propellane is initiated through metal carbenes, followed by nucleophilic addition of THF or alkynes. Additionally, control experiments in ref. 25 demonstrated that using FeCl₂ leads to the formation of methylenecyclobutane dimer via a carbene pathway, indicating that the ring opening of [1.1.1]propellane can proceed through an iron carbene intermediate. In our control experiments, we observed that the use of Fe(acac)₂ and Fe(acac)₃ with [1.1.1]propellane in methanol solvent resulted in the synthesis of alcohol-inserted methylenecyclobutane. Based on these findings, we suggested that this could be a minor pathway for the generation of methylenecyclobutane in our system. However, as noted by the reviewer, the efficiency of carbene formation can vary significantly depending on the ligand and reaction conditions. This is supported by the work of Uchiyama et al. (*J. Am. Chem. Soc.* **2017**, *139*, 17791), who showed that even when using an Fe(II) catalyst, the choice of ligand, such as Fe(Pc), can result in efficient radical insertion rather than ring opening of [1.1.1]propellane. This observation likely explains the low conversion observed with Fe(acac)₂. In contrast to Fe(acac)₂, the use of FeCl₂ as a catalyst, which is known for its efficient formation of iron carbenes, resulted in a notably rapid ring opening of [1.1.1]propellane. This suggests that the iron carbene pathway can be promoted under these conditions. To provide further evidence for this, we have included control experiments using FeCl₂ in the revised supporting information.

As suggested, we also conducted DFT calculations using FeCl₂ as a catalyst, which was found to generate a moderate amount of methylenecyclobutane in our control experiments. The computational results indicated that all the steps involved in the iron carbene pathway have feasible energy barriers, suggesting that this mechanism might indeed contribute as a minor pathway in the ring opening of [1.1.1]propellane under these conditions.

Free energy profile of the carbene generation and alcohol insertion. Energies (kcal/mol) in methanol solvent. All calculations were conducted using DFT as implemented in the Gaussian 09 suite. Gas-phase geometry optimizations were performed using the B3LYP functional and the 6-31G(d,p) basis set for all light atoms (H, C, O, Cl), while the Fe atom was treated with the LANL2DZ basis set. The single-point energies of optimized structures were re-evaluated with the B3LYP functional using the 6-311G(d,p) basis set for all light atoms (H, C, O, Cl) and the SDD basis set for the Fe atom. Solvation energy corrections were carried out at the same level as the single-point energy calculations using the SMD model (solvent = methanol).

- Another question deals with the Table S2: in the optimization of the Markovnikov addition the authors investigated different amounts of iron catalyst (5-20 mol%) without any effect on the yield. However, increase in alkene does raise the yield. What is the explanation for that?

Response and Action: Thank you for your insightful comments. Based on the observation of the reduced side product in the reaction, it is likely that the alkyl radical generated after MHAT competes between inserting into the *N*-amidopyridinium salt and undergoing HAT with the Si-H bond of the excess silane. Additionally, the formation of other side products, such as those from amidyl radical insertion and silane radical insertion (yielding <5%), results in a small consumption of the alkene. Therefore, even if we increase the amount of catalyst to promote the MHAT process, we cannot prevent the formation of side products, leading to a similar yield. However, by using an excess amount of alkene, the main reaction can proceed further despite the minor consumption of alkene via side pathways, resulting in an increase in yield.

- The representative procedure given in the manuscript should describe a more practical size scale (at least 0.5-1 mmol) and include all the amounts in mg of the reagents and substrates instead of only give the product amount.

Response and Action: We thank the reviewer for pointing this out. We conducted the 1,3-difunctionalization of [1.1.1]propellane and hydroxydation of alkene on a practical scale (1.0 mmol) and included the results in the scope and representative procedure sections in the revised manuscript and supporting information. The amounts of all reagents and solvent used were also provided.

Representative procedure for the 1,3-difunctionalization of [1.1.1]propellane. To a 50 mL round-bottom flask equipped with a magnetic bar, *N*-amidopyridinium salt **2a** (1.0 equiv, 1.0 mmol, 426.1 mg), sodium bicarbonate (1.0 equiv, 1.0 mmol, 84.0 mg), and iron catalyst Fe(acac)₃ (5 mol%, 0.05 mmol, 17.7 mg) were added. The flask was then sealed with a rubber septum, evacuated, and back-filled with argon. EtOH solvent (0.1 M, 10.0 mL) and Ph(OⁱPr)SiH₂ (2.0 equiv, 2.0 mmol, 0.36 mL) were added, followed by [1.1.1]propellane **1** (1.0 M in benzene, 1.5 equiv, 1.5 mmol, 1.5 mL) via syringe. The reaction mixture was stirred at room temperature for 24 h. After reaction completion, the reaction mixture was

diluted with ethyl acetate, washed with water, and extracted ethyl acetate three times. After removal of solvent, the residue was purified by flash column chromatography on silica gel (eluent: ethyl acetate/*n*-hexane = 1:10) to give the desired product **3a** (71%, 189.1 mg).

Representative procedure for the hydropyridylation of alkene. To a 50 mL round-bottom flask equipped with a magnetic bar, *N*-amidopyridinium salt **2a** (1.0 equiv, 1.0 mmol, 426.1 mg), sodium bicarbonate (1.0 equiv, 1.0 mmol, 84.0 mg), and iron catalyst Fe(acac)₃ (5 mol%, 0.05 mmol, 17.7 mg) were added. The flask was then sealed with a rubber septum, evacuated, and back-filled with argon. EtOH solvent (0.1 M, 10.0 mL) and Ph(OⁱPr)SiH₂ (2.0 equiv, 2.0 mmol, 0.36 mL) were added, followed by 2-methyl-1-hexene **5a** (1.5 equiv, 1.5 mmol, 0.21 mL) via syringe. The reaction mixture was stirred at room temperature for 15 h. After reaction completion, the reaction mixture was diluted with ethyl acetate, washed with water, and extracted ethyl acetate three times. After removal of solvent, the residue was purified by flash column chromatography on silica gel (eluent: ethyl acetate/*n*-hexane = 1:10) to give the desired product **6a** (77%, 194.5 mg).

Reviewer #3 (Remarks to the Author):

Hong described iron-hydride catalyzed C–H alkylation of pyridines at the C4 position for the difunctionalization of [1.1.1]propellane and hydropyridylation of alkenes. This protocol is synthetic useful. The optimization of reaction conditions is systematically demonstrated. The substrate scope is well studied. The catalytic pathway is well designed. The experimental procedures and characterization data are clean and well organized. In my opinion, this manuscript could be published in Nature Commun after the following minor revisions have been addressed.

Response and Action: We thank the reviewer for their careful evaluation of our work and their supportive feedback.

-The title, introduction section and Fig. 1 confused me. Although the sections show the reactions “the difunctionalization of [1.1.1]propellane and hydropyridylation of alkenes”, it is hard to grasp the essence. Please rewrite the sections. Which reaction is more important in this manuscript?

Response and Action: Thank you for your valuable feedback. Following the suggestion, we have revised the title to "1,3-Difunctionalization of [1.1.1]Propellane through Iron-Hydride Catalyzed Hydropyridylation" to better reflect the focus of our manuscript. Accordingly, we have updated the content of abstract and the introduction section to clarify the essence of the reactions discussed. We emphasize that the distinctive 1,3-difunctionalization of [1.1.1]propellane for the synthesis of an array of 1,3-difunctionalized cyclobutanes is the primary focus of our study, with the Fe-catalyzed Markovnikov hydropyridylation of various alkenes under mild reaction conditions serving as another significant aspect. We hope these revisions address your concerns and improve the clarity of our manuscript.

- Fig 4, 6ar-6aw; Is the regioisomer observed? If not, the author should discuss the reason.

Response and Action: We thank the reviewer for pointing this out. Regarding the regioselectivity, we observed no regioisomers for each substrate. For compound **6ar**, we initially hypothesized that the boronic ester could stabilize the radical through delocalization, potentially leading to the formation of the proximal selective product. However, our experimental results showed distal selectivity. Although the exact reason for this selectivity is difficult to explain at this point, similar distal selectivity has been reported in previous studies involving pinacol boronic ester MHAT (ref. 76). For compounds **6as-6aw**, the regioselectivity of MHAT is determined by the preference of the alkyl radical intermediate, both thermodynamically and kinetically. In this context, the tertiary alkyl radical is favored, leading to β -selectivity. Furthermore, since the carbonyl α radical is electrophilic, it is less likely to insert into the electron-deficient pyridinium salt. Consequently, the tertiary alkyl radical exhibits remarkable selectivity.

- Table 1, Can the author decrease the amount of alcohol? How is about other nucleophiles such as phenol or amines?

Response and Action: We thank the reviewer for pointing this out. In the case of the 1,3-difunctionalization of [1.1.1]propellane, a rapid ring opening is crucial to achieve high selectivity in the functionalization process. If the equivalents of alcohol are reduced, the ring opening of [1.1.1]propellane does not proceed quickly enough, leading to an increased formation of product **B**, which results from direct MHAT, as shown below. As suggested, we attempted using other nucleophiles, including phenol and amines, but most of them showed only trace reactivity as detected by LCMS. Only when aniline (4.0 equiv) was used as the nucleophile, we were able to obtain 6% of the desired product. Increasing the amount of aniline to 10.0 equivalents still resulted in low reactivity.

Entry	Solvent	variation	Yield of A (%)	Yield of B (%)
1	EtOH : MeCN = 1 : 1	-	31	16
2	EtOH : MeCN = 3 : 1	-	38	10
3	EtOH : MeCN = 9 : 1	-	50	7
4	EtOH	-	73	trace
5	MeCN	EtOH (4.0 equiv) additive	<5%	16
6	EA	EtOH (4.0 equiv) additive	<5%	11
7	DCM	EtOH (4.0 equiv) additive	<5%	6

a : Using 10 equiv of nucleophile

- Fig 3, The use of bulky alcohol is nice. However, the yield of bulky alcohol is not excellent. Can the author improve it?

Response and Action: We thank the reviewer for pointing this out. When using a bulky alcohol solvent, we hypothesized that the low yield could be attributed to the reduced nucleophilicity of the bulky alcohol during the nucleophilic attack process. To address this issue, we attempted to promote the nucleophilic addition process by heating the reaction. However, similar reactivity was observed at elevated temperatures, and a significant decrease in reactivity was noted at temperatures above 80°C due to the occurrence of side reactions. There is a possibility that during the MHAT process, bulky alcohols may not efficiently undergo hydride/alkoxy exchange during the generation of metal hydrides. To investigate this, we attempted to use ethanol as an additive; however, it still exhibited low reactivity.

Entry	Temperature	additive	Yield (%)
1	25 °C		43
2	40 °C		42
3	60 °C		39
4	80 °C		20
5	25 °C	EtOH 4.0 equiv	44

Reviewers' Comments:

Reviewer #2:

Remarks to the Author:

The revised form of the manuscript submitted by Hong et al. has clarified a number of things raised during the previous review process and made the manuscript better understandable. They also included a larger scale procedure for the reaction.

The authors did not mention the possible availability of the silane Ph(OiPr)SiH₂ they use, as it is not really commercially available. Did they investigate other silanes except PhSiH₃ and PhMeSiH₂? They are not reporting the investigation of the latter silanes under the final optimized conditions, which would be interesting. The possibility of using commercial available silanes for the reaction would definitely lower the barrier for application in other syntheses.

It is also still hard to understand, why a change in the amount of catalyst has such a small influence on the reaction outcome and also why there is an optimum for the amount of alkenes (1.5 equiv.). The independence from the catalyst would mean that the course of the reaction is rather independent from the catalyst amount, which is hard to grasp for a catalytic reaction, when the precatalyst is involved in a important/rate-determining step of the reaction. According to the authors data the catalyst amount has basically zero influence (what happens with less catalyst then?). This point appears to be quite important and the explanation.

Reviewer #3:

Remarks to the Author:

The authors have addressed most of the issues raised by the reviewers. I recommend it for Nat. Commun.

Reviewer #2 (Remarks to the Author):

The revised form of the manuscript submitted by Hong et al. has clarified a number of things raised during the previous review process and made the manuscript better understandable. They also included a larger scale procedure for the reaction.

Response and Action: We thank the reviewer for their careful evaluation of our work and their supportive feedback.

- The authors did not mention the possible availability of the silane Ph(OiPr)SiH₂ they use, as it is not really commercially available. Did they investigate other silanes except PhSiH₃ and PhMeSiH₂? They are not reporting the investigation of the latter silanes under the final optimized conditions, which would be interesting. The possibility of using commercial available silanes for the reaction would definitely lower the barrier for application in other syntheses.

Response and Action: We thank the reviewer for pointing this out. The Ph(OⁱPr)SiH₂ used in the reaction is commercially available and was purchased from Sigma-Aldrich (catalog number: 900258). During the optimization step, we conducted silane screening with NaOAc base, and tri-substituted silanes such as Ph₃SiH, Et₃SiH, (EtO)₂MeSiH, (EtO)₃SiH, and (iPr)₃SiH showed only trace or no reactivity. Therefore, we retried the reaction under the final optimized conditions with mono-, di-, and tri-substituted silanes. However, all of them showed reduced reactivity compared to Ph(OⁱPr)SiH₂. The silane screening data has been added to the revised supplementary information.

Entry	Silane	Yield (%)
1	PhSiH ₃	58
2	PhMeSiH ₂	6
3	Ph ₃ SiH	trace
4	(EtO) ₂ MeSiH	13

Entry	Silane	Yield (%)
1	PhSiH ₃	23 (d.r. = 2.8 : 1)
2	PhMeSiH ₂	4
3	Ph ₃ SiH	trace
4	(EtO) ₂ MeSiH	12 (d.r. = 2.9 : 1)

- It is also still hard to understand, why a change in the amount of catalyst has such a small influence on the reaction outcome and also why there is an optimum for the amount of alkenes (1.5 equiv.). The independence from the catalyst would mean that the course of the reaction is rather independent from the catalyst amount, which is hard to grasp for a catalytic reaction, when the precatalyst is involved in a important/rate-determining step of the reaction. According to the authors data the catalyst amount has basically zero influence (what happens with less catalyst then?). This point appears to be quite important and the explanation.

Response and Action: We apologize for any confusion caused by the result for a single substrate. In our study, there are some substrates in the scope where 10 mol% Fe was used to increase reactivity and yields. In these cases, employing 5 mol% Fe led to reduced yields due to an insufficient reaction rate or side reactions, and a higher reactivity was observed when 10 mol% Fe was utilized. To prevent confusion, we have added these results in the revised supplementary information Figure S1.

Regarding the amount of alkenes, no significant advantages were observed when using more than 1.5 equivalents. In Table S2 of the supplementary information, yield analysis was conducted using ¹H NMR. In the case of the Fe catalyst, its paramagnetic properties tend to cause a slight broadening of the overall NMR peak, even after work-up. This can result in a slight difference between the isolation yield and the ¹H NMR yield in the optimized table. Therefore, the 3% yield difference observed in entries 6 (alkene 1.5 equiv) and 7 (alkene 2.0 equiv) is considered within the margin of error. Using 2.0 equivalents of alkene did not lead to any meaningful increase in reactivity, so we decided to utilize 1.5 equiv of alkenes as the optimized condition.

Reviewer #3 (Remarks to the Author):

The authors have addressed most of the issues raised by the reviewers. I recommend it for Nat. Commun.

Response and Action: We thank the reviewer for their support of our work.